# Implications of Artificial Intelligence Algorithms in the Diagnosis and Treatment of Motor Neuron Diseases—A Review

**DOI:** 10.3390/life13041031

**Published:** 2023-04-17

**Authors:** Diego Lopez-Bernal, David Balderas, Pedro Ponce, Mario Rojas, Arturo Molina

**Affiliations:** Tecnologico de Monterrey, National Department of Research, Puente 222, Del. Tlalpan, Mexico City 14380, Mexico; lopezbernal.d@tec.mx (D.L.-B.);

**Keywords:** motor neuron diseases, artificial intelligence, diagnosis, treatment, prognosis

## Abstract

Motor neuron diseases (MNDs) are a group of chronic neurological disorders characterized by the progressive failure of the motor system. Currently, these disorders do not have a definitive treatment; therefore, it is of huge importance to propose new and more advanced diagnoses and treatment options for MNDs. Nowadays, artificial intelligence is being applied to solve several real-life problems in different areas, including healthcare. It has shown great potential to accelerate the understanding and management of many health disorders, including neurological ones. Therefore, the main objective of this work is to offer a review of the most important research that has been done on the application of artificial intelligence models for analyzing motor disorders. This review includes a general description of the most commonly used AI algorithms and their usage in MND diagnosis, prognosis, and treatment. Finally, we highlight the main issues that must be overcome to take full advantage of what AI can offer us when dealing with MNDs.

## 1. Introduction

Motor neuron diseases (MNDs) are a group of neurological disorders that are characterized by the progressive dysfunction of motor neurons (MNs). These motor neurons can be divided into two groups based on their location within the central nervous system (CNS): lower motor neurons and upper motor neurons. The lower MNs are located in the brainstem and spinal cord, while the upper MNs are located in the motor cortex [1]. Motor neurons regulate the contraction of skeletal muscles; therefore, when the motor nerve axon degenerates, the muscle fibers that are innervated by those neurons will stop working properly, and the patient will start losing motor control. Some of the most important MNDs include amyotrophic lateral sclerosis (ALS), spastic paraplegia, bulbar palsy, polytopic paralysis, primary lateral sclerosis (PLS), post-polio syndrome, and spinal muscular atrophy.

Of all the previously mentioned disorders, ALS is the most common, often resulting in death within 3–5 years after diagnosis [2]. However, other MNDs also pose a global problem, as their poor long-term prognosis has caused a significant socioeconomic burden worldwide [3]. In 2019, motor neuron disorders caused 39,081 deaths and 1,034,606 disability-adjusted life years (DALYs) [3]. To decrease the global burden that MNDs represent, healthcare providers must improve their diagnostic and management methods, so that patients can receive the best treatment at the best possible time.

In absence of an effective treatment for MNDs, early diagnosis is a highly relevant, yet challenging task. The main reason for this is that there is high clinical heterogeneity and no definitive biomarkers for different MNDs. These factors often contribute to a diagnostic delay that can go from 9 months up to 27 months [4,5]. Moreover, the uncertainty during MND detection can also cause a patient to be subject to unneeded interventions [6,7], which can accelerate the progression of the disease [8] and even increase the diagnosis time [9,10,11]. Therefore, to improve diagnosis times and decrease the impacts on patients, medical providers must find an optimum way to analyze MND clinical data that allows for correct disease stratification. Here is where artificial intelligence (AI) models may become handy and useful.

In healthcare, artificial intelligence has been used for medical data and image analysis, health monitoring, disease diagnosis, drug discovery, and the development of new protocols. For example, an AI-enabled platform named AliveCor Kardia has been proven to successfully identify atrial fibrillation while being well-accepted by the majority of patients [12]. Moreover, in cardiology, several machine learning algorithms, such as support vector machine (SVM), random forest (RF), and logistic regression (LR) have been used to predict major adverse cardiac events (MACE) of acute coronary syndrome (ACS) [13] and predict readmission after hospitalization due to heart failure [14]. Moreover, AI algorithms have been applied to diabetes care. Some of these applications include Healthcare Recommendation System(s) (HRS) for diabetes risk and complication predictions [15,16,17], automated retinal screening for diabetic retinopathy diagnosis [18], and short–long-term glycated hemoglobin response monitoring after insulin initiation [19]. Another specialty that has benefited from AI is gastroenterology by using algorithms for the detection of malignant and premalignant lesions, Helicobacter pylori diagnosis, gastrointestinal bleeding identification, celiac disease identification, etc. [20].

As can be seen, AI applications in healthcare are rapidly evolving and their benefits are undoubtedly evident. Different algorithms have proven their usefulness to enhance the detection procedures for different medical pathologies. Therefore, the main objective of this work is to provide a brief introduction to some of the most common artificial intelligence algorithms and their current clinical applications, limitations, and future directions in diagnosing and treating motor neuron diseases.

## 2. Artificial Intelligence

Artificial intelligence is a branch of computer science that aims to replicate the cognitive functions of humans such as learning and problem-solving [21]. The foundation of AI is called machine learning (ML), which can be described as a set of mathematical models trained using a dataset to predict or execute a determined action [22]. Machine learning can be divided into three main branches: supervised learning, unsupervised learning, and reinforcement learning. Supervised learning requires labeled data to learn from, while unsupervised learning aims to find patterns in the data and learn from them without needing labeled data. Reinforcement learning involves machine learning from “punishments” and “rewards”. In healthcare applications, supervised learning is the most commonly used learning method, followed by unsupervised learning. The following sections provide a brief and general description of the supervised and unsupervised algorithms most frequently used in the diagnosis and treatment of motor neuron diseases.

### 2.1. Support Vector Machines

A support vector machine, also known as SVM, is a supervised learning method that searches how to classify data by maximizing the difference between each category, making it easier to distinguish between each of the classes (green and red circles). This is done using hyperplanes and support vectors, as shown in Figure 1. Support vector machines are used for binary and multi-class classification [23].

### 2.2. Random Forest

Random forest is a supervised classifier that consists of a collection of decision trees working together as an ensemble for supervised learning. A decision tree is a hierarchically organized algorithm that can classify or predict an outcome based on a set of features and rules [24]. An example of this algorithm is shown in Figure 2.

Then, in a random forest, each decision tree has a unit vote so that the final classification or prediction is given by the most voted outcome, as shown in Figure 3.

### 2.3. K-Nearest Neighbor

K-nearest neighbor (KNN) is a supervised algorithm that classifies a new observation based on its similarity to previously stored data [25]. To classify a new data point, the algorithm searches for the labels of its nearest neighbors (NNs) so that the label that appears the most among the NNs is assigned to the new observation. The final classification of the algorithm depends on the number of nearest neighbors *k*. For example, in Figure 4, the new data point is classified as class 2 when k=3 and as class 1 when k=7. Therefore, to obtain an accurate classification, the algorithm must be tested with different *k* values to find the most optimum one for the given classification task.

### 2.4. Linear Discriminant Analysis

Linear discriminant analysis (LDA) is a supervised algorithm used to find the most important and representative characteristics of a group and classify subjects into two different categories. This is done by maximizing the between-class variance and minimizing the within-class variance [26]. For example, in Figure 5, we have red and green dots (classes) scattered around two independent variables, as illustrated by the X1 axis and X2 axis.

From here, the main objective is to obtain a new axis to project the data to better separate both classes. Figure 6 and Figure 7 represent two different ways in which we can create the new axis. However, Figure 6 shows an axis that maximizes between-class variance (arrow between means) but ignores within-class variance (dotted ellipses). On the other hand, Figure 7 represents an axis that achieves both: maximizes between-class variance and minimizes within-class variance, achieving a good class separation.

### 2.5. Naive Bayes

Naïve Bayes is a supervised classifier based on the naïve Bayes theorem, which is used to solve classification problems based on a determined number of properties $p_{i}$ and the probability of subjects $x_{i}$ having those properties [27].

### 2.6. Principal Component Analysis

Principal component analysis (PCA) is an unsupervised dimensionality reduction technique. It is normally used for exploratory data analyses or to find a relation between a given set of variables [28]. The main goal of PCA is to transform a large set of variables into a smaller one without compromising so much information. In this way, the information can be analyzed easier and faster.

### 2.7. Linear Regression

Regression models are used as supervised algorithms in machine learning to predict a dependent variable *y* given some independent variables *x* [29]. The most common linear regression models are:Simple linear regression (LR): consists of a model in which the main objective is to search how a dependent variable (*y*) is related to the independent variable (*x*).Multivariate linear regression (MLR): this model is used to measure the relationship between a dependent variable (*y*) and a set of diverse independent variables ($x_{1},x_{2},x_{3},...,x_{n}$).

### 2.8. Artificial Neural Networks

Artificial neural networks, also known as ANNs, are machine learning models that can be either supervised or unsupervised, inspired by the way the human brain works. The main objective of these models is to mimic the way real neurons work using mathematical operations. This type of algorithm is more complex than the previous one, however, it can lead to better results because of its high capacity to analyze different variables at the same time. Nevertheless, one of the main issues is that ANNs need a lot of data to learn from to work correctly. If there is not enough data to train it, the ANN can generate a wrong final classification. As shown in Figure 8, the training data will enter the input layer, then it is processed in the hidden layers so that the ANN learns from it; finally, the classification is shown in the output layer.

### 2.9. Convolutional Neural Networks

Convolutional neural networks (CNNs), as shown in Figure 9, are a special type of artificial neural network typically used for image analysis and computer vision [30]. This type of network takes an image as input and assigns a weight and bias to each pixel. These weights and biases are adjusted through a learning process. A typical CNN consists of the following operations: input, convolution, pooling, flattening, fully connected layer, and output. A detailed description of each of these operations is beyond the scope of this work, but more information can be found in [31].

### 2.10. Human–Computer Interfaces

Human–computer interfaces (HCIs) are not machine learning algorithms, but rather a technology that utilizes those algorithms to function properly. This technology is based on the study of the human body as a means of controlling an electronic device, such as a computer [32,33]. Different human actions can serve as control signals in HCI, such as eye movement [34], tongue movement [35], body motion [36], and brain/muscle electrical signals [37], among others. Therefore, this technology offers several benefits when treating motor neuron diseases, as will be mentioned later in this paper.

## 3. Machine Learning and HCI in Motor Neuron Diseases

The previously mentioned machine learning algorithms and tools have been widely applied in the context of motor neuron diseases. Mainly, ML algorithms are used for the diagnosis, prognosis, and monitoring of MNDs, while HCIs are used as a "treatment" option for MND patients to recover some movement capabilities.

### 3.1. MND Diagnosis Using ML Algorithms

Timely diagnosis of motor neuron diseases is essential for improving the likelihood of having a good quality of life and survival rate, while also decreasing the social burden that MNDs represent. Therefore, many research studies have focused on applying machine learning algorithms for the early detection of motor neuron diseases.

#### 3.1.1. ALS Diagnosis

In work by Sarica et al. [38], diffusion tensor imaging (DTI) was used to obtain different metrics for binary classification of twenty-four ALS patients v.s. twenty-four healthy control subjects. The classification was conducted through a random forest algorithm, achieving a mean accuracy of 80%, with a maximum of 87.5% and a minimum of 75%. The RF algorithm was also applied by Ferraro et al. [39] to DTI metrics and cortical thickness values for multi-class classification of patients with ALS, predominantly upper motor neuron disease (PUMN), ALS-mimic disorders, and healthy controls, leading to a mean classification accuracy of 84.6%. Moreover, the study by Fratello et al. presented a multi-class classification of ALS patients, Parkinson’s patients, and healthy controls, obtaining a maximum accuracy of 82.9% using random forest [40]. Moreover, Thome et al. applied an RF algorithm to classify healthy controls and ALS patients using brain volume metrics and resting state functional connectivity metrics, achieving a mean accuracy of 66.8% [41]. Apart from RF, SVM is another algorithm that has been widely applied for ALS and other MND diagnoses.

The work by D’hulst et al. demonstrates the use of SVM to classify ALS patients versus healthy controls using positron emission tomography metrics [42]. This study obtained a maximum classification accuracy of 94.8%. and a minimum of 12.5%. Furthermore, Costagli et al. carried out a binary classification of ALS patients v.s. healthy controls using different statistical metrics of the iron concentration increase in the primary motor cortex of ALS subjects [43]. The maximum accuracy achieved by this study was 90%. Moreover, support vector machines were used by Chang et al. to classify ALS patients and healthy subjects using plasma biomarkers metrics, achieving a mean accuracy of 88% [44]. Additionally, the research presented by Perez-Ortiz et al. showed that EEG activity monitoring, coupled with a support vector machine classifier, can also be used for ALS diagnosis [45], leading to a success rate of 89%. Research by Bandini et al. was conducted to distinguish between presymptomatic and symptomatic ALS stages using SVM and KNN algorithms [46]. The mean accuracy for SVM was 85.3% and for KNN was 83.8%.

The K-nearest neighbor algorithm, along with other algorithms (such as RF, DT, and NB) was also used in research conducted by Quintão et al. to diagnose ALS based on surface electromyography dynamical features [47]. The top accuracy in this study was 94% with an average of 77%. Another algorithm that has been used to diagnose amyotrophic lateral sclerosis is LDA. Likhachov et al. used LDA to evaluate the voice function state of patients with ALS to estimate if the current state is pathological or if it represents a risk of disease progression [48]. Furthermore, convolutional neural networks have been applied to ALS diagnoses. The work by Imamura et al. [49] shows a prediction model in which CNNs are used to diagnose ALS through the analysis of images of motor neurons. The average classification accuracy obtained in this study was 90%.

#### 3.1.2. Other MND Diagnoses

Despite most of the work being directed toward ALS, some studies have been done to test machine learning algorithms in the diagnosis of other motor neuron diseases. For example, the research done by MacWilliams et al. used a decision tree model denominated BART (Bayesian additive regression tree) to discriminate between hereditary spastic paraplegia and cerebral palsy [50]. Moreover, Sekar et al. proposed an algorithm called neuro machine learning model to predict several MNDs such as bulbar palsy, polytopic paralysis, and ALS [51]. Furthermore, Bede et al. provided a model of an artificial neural network to differentiate patients with ALS, primary lateral sclerosis, and post-polio syndrome [52]. In this work, the greatest accuracy was obtained for ALS (93.7%), followed by post-polio syndrome (60.3%), and primary lateral sclerosis (43.8%). Moreover, regression models have been used for the diagnosis of spinal muscular atrophy, obtaining a high prediction accuracy, as reported by Chen et al. [53].

### 3.2. MND Prognosis and Monitoring Using ML Algorithms

The main objective of prognosis and monitoring of any disease is to optimize treatment strategies to decrease the risk of the disease worsening or being fatal. Most of the prognosis and monitoring applications of machine learning have been directed toward ALS. Several works, such as those presented by Hothorn et al., Ko et al., and Taylor et al., trained random forest algorithms using electronic health record data, such as demographics, family history, medical history, etc., to predict ALS disease progression [54,55,56].

Another study presented by Huang et al. showed a survival analysis of ALS patients using random forest [57]. Furthermore, the research by Beaulieu-Jones et al. presented how RF, LR, SVM, and KNN algorithms can be used to predict the life expectancy (number of days) of ALS patients from disease onset [58]. Moreover, predictive regression models and RF were used by Pfohl et al. to identify the clinical ALS metrics that are most helpful in predicting the survival rate of patients and how their predictive ability changes as the disease progresses [59]. Linear discriminant analysis has also been applied to ALS prognosis by analyzing the state of the voice of patients [48]. Furthermore, artificial neural networks have been used to analyze ALS progression and survival rate. For example, van der Burgh et al. trained different ANNs using clinical data and magnetic resonance imaging to predict survival in ALS patients [60]. A study by Pancotti et al. presented three different artificial neural network architectures for ALS progression prediction [61].

### 3.3. Machine Learning Algorithms to Assist Patients with MND

To date, there is generally no effective treatment for motor neuron diseases. Therefore, machine learning has been used as an assisting option for patients with MND. Most of these options are based on human–computer interfaces that can help patients recover some movement abilities despite ALS progression. It is important to remember that, most of the time, motor neuron diseases can lead the patient to move into a locked-in state, in which the subject is not able to move on their own. A particular type of HCI known as the brain–computer interface (BCI) is useful in these situations. This type of interface does not need any type of movement because it focuses on the analysis of brain activity, which usually remains intact even in subjects with MND [62,63].

There have been several works with the main objective of researching the application of BCI toward the rehabilitation of MND patients. For example, the work presented by de Oliveira Junior et al. [64] proposes a Smart Home device to help ALS patients perform given tasks. This is accomplished through a BCI system called ALSHelp, which consists of a brainwave wearable sensor and a mobile application. The screen of the mobile application shows images representing the tasks that can be executed. Then, the application periodically changes from task to task, waiting for the user to select one of them. To select it, the user must blink an eye so that the system detects this action and performs the selected task. Moreover, Hosni et al. presented a motor imagery BCI based on functional near-infrared spectroscopy [65]. This BCI system used a support vector machine classifier to discriminate between a motor imagery state and a rest state in ALS patients. Furthermore, research by Guy et al. presents a BCI speller based on EEG signals for patients with ALS [66]. Another study explored a BCI for a communication system based on eye-tracking and visual stimulus [67]. These works are just examples of BCI applications in ALS patients. According to Vaughan et al., as of 2019, 110 peer-reviewed articles on BCI systems included ALS patients as study subjects [68]. Therefore, it can be presumed that there is notable research interest in using BCI as a non-pharmaceutical treatment option for patients with MND.

## 4. Limitations

Despite machine learning showing good performance towards diagnosis, prognosis, and treatment of MND, it still has several limitations that it must overcome. In general, ML has been shown to face the following challenges when being applied to healthcare:Missing data, sparsity, and biasing: machine learning models need a large amount of data to be trained properly. In healthcare, there is usually a lack of data because of the cost of acquiring information, the difficulty of the procedures, the time consumed during data acquisition, ethical issues, patients not going to all of their check-ups, etc. Moreover, the data obtained during clinical procedures or medical check-ups are usually not clean or well structured. Therefore, it is not rare to find noisy data and redundant values in healthcare datasets [69].Complexity: biological systems are not easy to understand. There are a lot of variables to take into account when attempting to analyze the human body and how it responds to a given stimulus or state. Therefore, the complexity of healthcare data can be challenging for ML applications in medicine [70].Interpretability: machine learning algorithms are usually treated as black boxes, meaning that there is no true understanding of how and why the algorithm works. This is an issue because medical professionals and healthcare providers need to interpret the recommended actions from an ML model for them to apply those actions to a patient [71].Operational challenges: most of the time, ML applications need patients to be followed for long periods to obtain enough data. Moreover, constructing and deploying the model requires personnel with adequate skills to develop it and understand the obtained results. Furthermore, ML-driven systems must be easy to integrate into the clinical workflow.Ethical issues: this is currently a huge issue for ML applications in healthcare. Despite machine learning models being useful tools, they are not perfect and can make mistakes. In medicine, this may cause healthcare providers to be misled and make incorrect decisions. Moreover, all of the information used in ML models must be anonymized due to data ownership and patient privacy [72]. Moreover, transparency standards must be satisfied. Therefore, machine learning models cannot be non-interpretable [73].Overfitting: this is a very common issue with machine learning models. It means that the model works correctly when dealing with a given set of data, but fails to generalize its results when applied to a different set of data. This is a problem in healthcare because models cannot be trained for every single patient and we must guarantee the correct performance in cases where those models are applied [74].

## 5. Conclusions and Future Directions

Machine learning has become a useful tool for several healthcare applications, including diagnosis, prognosis, and treatment of motor neuron diseases, such as ALS, spastic paraplegia, bulbar palsy, polytopic paralysis, primary lateral sclerosis, and post-polio syndrome. However, it is important to note that most of the research on ML applied to MND has focused on ALS, as it is the most common and important motor neuron disease. Moreover, brain–computer interfaces have also been widely proposed and used to offer MND patients with new motor capabilities despite being in a locked-in state. However, both ML and BCI still face challenges in becoming fully applicable in real-life situations. Most of these novel technologies are considered black boxes, which are difficult to interpret and have limited generalizability. These issues are mainly caused by problems such as missing data, data sparsity, bias, and overfitting. Additionally, researchers must be careful when dealing with the medical data of patients to avoid violating their privacy and security rights. Therefore, more research is needed to understand how to adequately tackle these issues and take full advantage of the benefits that ML may bring to motor neuron disease diagnosis, prognosis, and treatment.

## Figures and Tables

**Figure 1 life-13-01031-f001:**
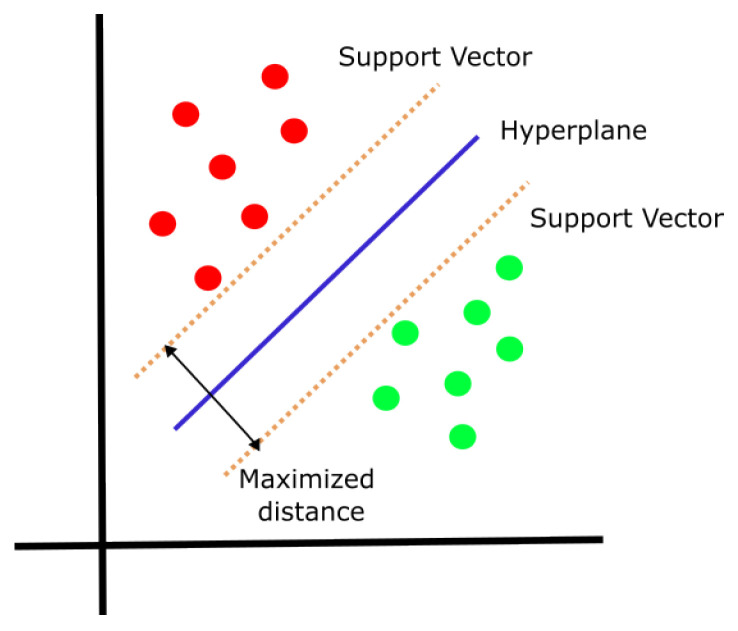
General representation of support vector machines.

**Figure 2 life-13-01031-f002:**
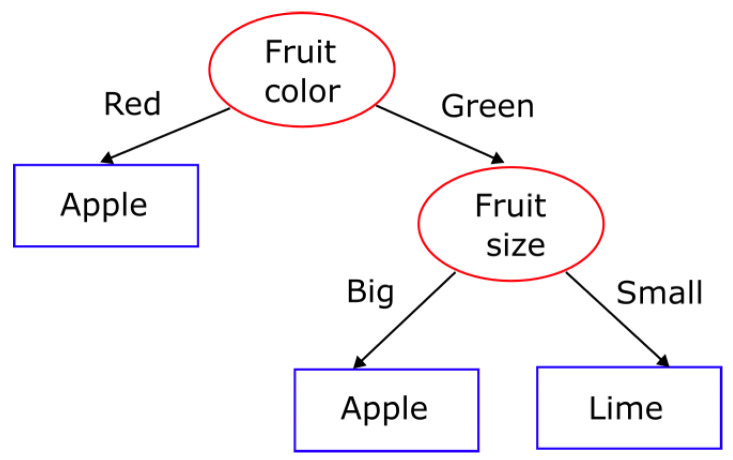
Decision tree.

**Figure 3 life-13-01031-f003:**
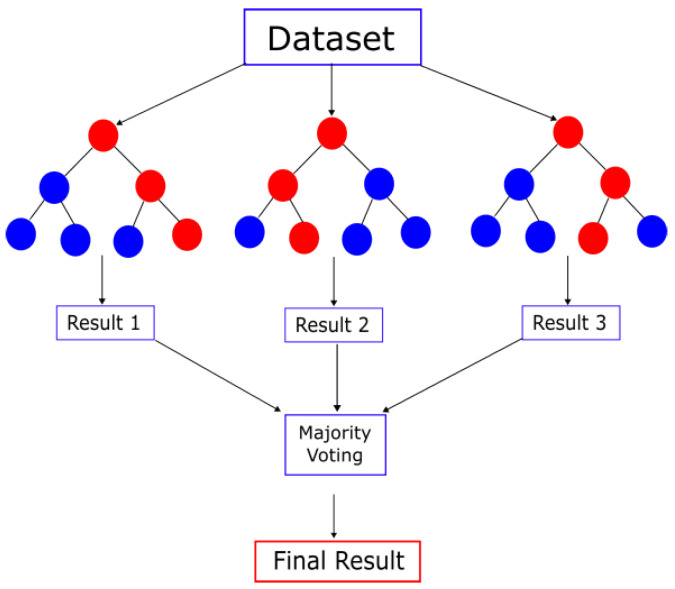
Random forest.

**Figure 4 life-13-01031-f004:**
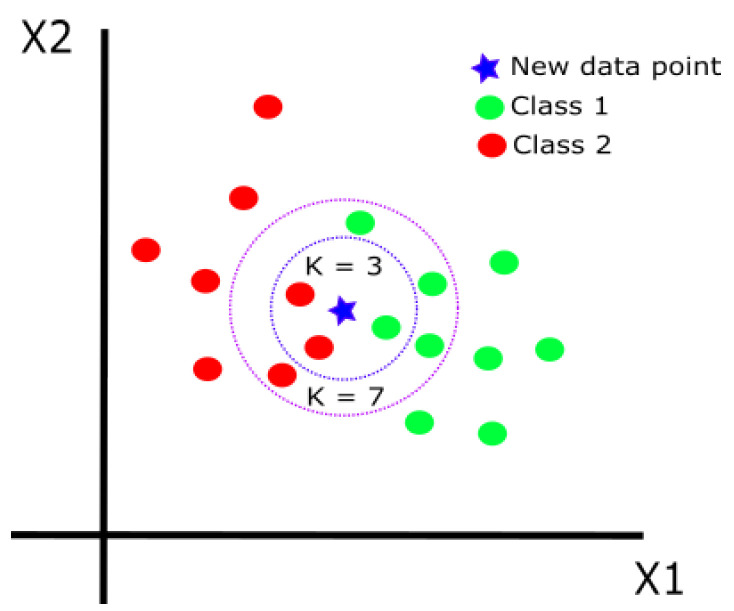
K-nearest neighbor.

**Figure 5 life-13-01031-f005:**
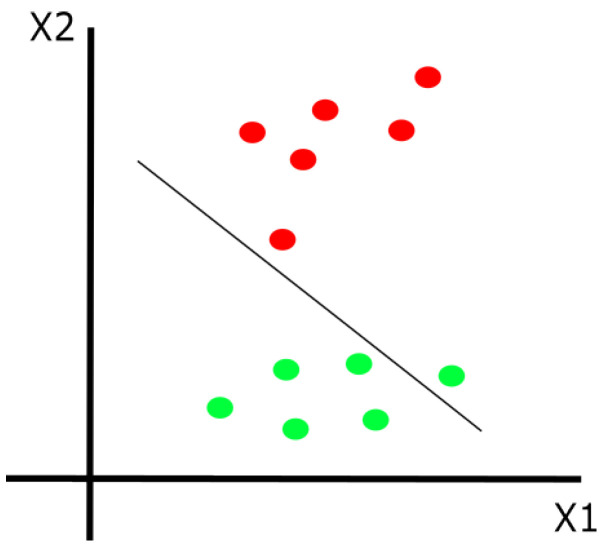
The 2D distribution of variables.

**Figure 6 life-13-01031-f006:**
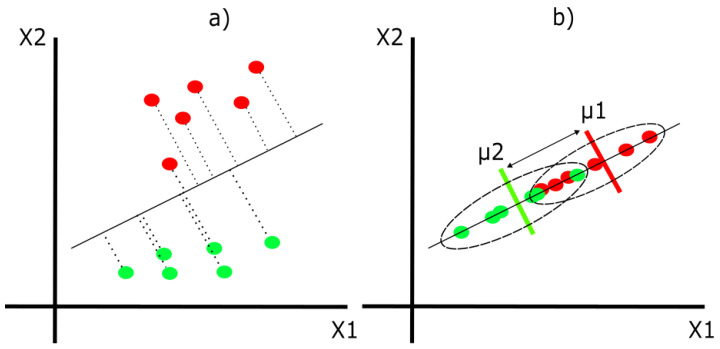
(**a**) The projection of data points on a new “bad” axis. (**b**) The resulting separation with the new axis.

**Figure 7 life-13-01031-f007:**
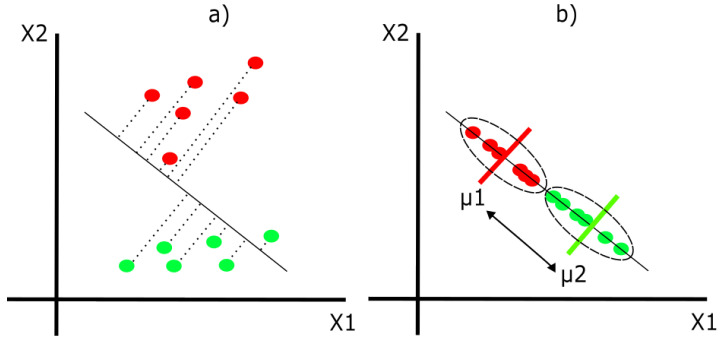
(**a**) The projection of data points on a new “good” axis. (**b**) The resulting separation with the new axis.

**Figure 8 life-13-01031-f008:**
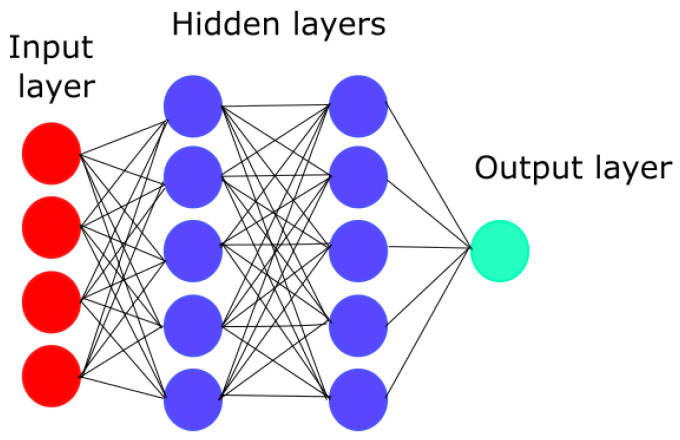
Artificial neural network model.

**Figure 9 life-13-01031-f009:**
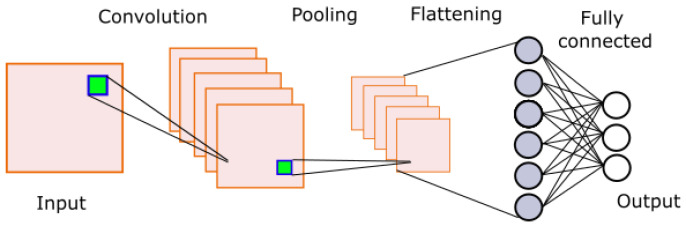
Convolutional neural network model.

## Data Availability

No new data were created or analyzed in this study. Data sharing is not applicable to this article.

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
