# Peer review of "Implications of Artificial Intelligence Algorithms in the Diagnosis and Treatment of Motor Neuron Diseases—A Review"

_life, 2023, doi:10.3390/life13041031_

Round 1

Reviewer 1 Report

Dear authors, i read with interest your manuscript "Artificial Intelligence algorithms as a nonpharmaceutical diagnosis and treatment option for motor neuron diseases"

ALS is a fatal disease without effective treatment option to date. The early diagnosis is essential in order for potential treatment.

I have some major concerns 

The title should be changed since diagnosis cannot be "pharmaceutical" and artificial intelligence cannot treat MNDs. You are reporting a narrative review describing the implications of algorithms-based approachs in the diagnosis and treatment of MND.  

1. Introduction is confusing and it does not introduce the purpose of the review. Also, it is enriched of unnecessary information (e.g., information about other diseases). 

2. The section artificial intelligence provides a lot of technical information about machine learning algorithms (6/13 pages; about fifty percent of the manuscript) and this could be to heavy for not-mathematical readership. 

3)MND section, you reported the recent published paper in the field although the section is confusing. You cannot start the paragraph or a sentence citing "as in [42]" but you must cite the first author o the study type rather than the reference. This error is everywhere throughout the manuscript.

This manuscript could be a guide for the selection of appropriate machine learning approach in the case of a study design but it is not a review of "Intelligence algorithms as a nonpharmaceutical diagnosis and treatment option for motor neuron diseases" since the issues related to MND diagnosis are almost neglected. It seems a book chapter about machine learning approach rather than a scientific paper. 

Kind regards

Reviewer 2 Report

Manuscript was written well and it has a great originality and novelty.  It will be very useful to the society.  I recommend minor revision before publication.

1. Please check the grammatical error of the sentence in L91.

2. I suggest a english correction by professional english service.  So that every section gives proper sense. 

Reviewer 3 Report

In the manuscript entitied“Implications of artificial intelligence algorithms in the diagnosis and treatment of Motor Neuron Diseases - A review", the authors summarized application of artificial intelligence models for the motor neuron diseases (MNDs).  They first described the current situation of MNDs and introduced the most common artificial intelligence (AI) algorithms. Then they focus on the matchine learning algorithms and its application in MNDs diagnosis, prognosis and monitoring. The authors present limitations of ML in diagnosis and treatment of MNDs. Finally, they have conclusions and futrue directions for ML. Overall, the logic and content are good. It is likely to be of great interest to the scientists who study the analysis and diagonis of MNDs.